# The Role of Pharmacotherapy in Treatment of Meningioma: A Systematic Review

**DOI:** 10.3390/cancers15020483

**Published:** 2023-01-12

**Authors:** Ataollah Shahbandi, Darsh S. Shah, Caroline C. Hadley, Akash J. Patel

**Affiliations:** 1Department of Neurological Surgery, School of Medicine, Tehran University of Medical Sciences, Tehran 1416634793, Iran; 2Department of Neurological Surgery, Dell Medical School, Austin, TX 78712, USA; 3Department of Neurosurgery, Baylor College of Medicine, Houston, TX 77030, USA; 4Department of Otolaryngology-Head and Neck Surgery, Baylor College of Medicine, Houston, TX 77030, USA; 5Jan and Dan Duncan Neurological Research Institute, Texas Children’s Hospital, Houston, TX 77030, USA

**Keywords:** meningeal neoplasms, meningioma, clinical trial, antineoplastic protocols, antineoplastic agents, chemoradiotherapy, targeted inhibitors

## Abstract

**Simple Summary:**

For the last 35 years, various systemic therapies for recurrent or refractory meningiomas have been investigated. The present review aggregated the currently available evidence in the literature regarding the safety and efficacy of these treatments and assessed the ongoing trials of medical therapy for meningiomas. The findings of the present study would assist future research in seeing what therapeutic regimens have been investigated, which targets are promising candidates for interventions, and how the ongoing clinical trials are currently designed.

**Abstract:**

The safety and efficacy of various pharmacotherapeutic regimens on refractory meningiomas have been the focus of investigations. We present a comprehensive review of the previous efforts and the current state of ongoing clinical trials. A PRISMA-compliant review of the MEDLINE and ClinicalTrial.gov databases of the National Library of Medicine were performed. The primary outcomes of interest for included articles were radiographic response, overall survival, progression-free survival, six-month progression-free survival, and adverse events. Overall, 34 completed trials and 27 ongoing clinical trials were eligible. Six-month progression-free survival was reported in 6–100% of patients in the completed studies. Hematological disorders were the most common adverse events. Of the ongoing clinical trials identified, nine studies are phase I clinical trials, eleven are phase II trials, two are phase I and II trials, one is phase II and III, and two trials do not have a designated phase. Currently, there is no effective chemotherapy for refractory or recurrent meningiomas. Several promising targeted agents have been developed and are currently being investigated in the hope of identifying novel therapeutic strategies for the treatment of this pathology.

## 1. Introduction

Meningiomas are the most common primary tumor of the central nervous system, accounting for 38% of all brain and spinal cord tumors. The five-year prevalence is approximately 159,038 cases and the incidence is 8.81 annual cases per 100,000 people. Histologically, 80.3% of these tumors are classified by the World Health Organization (WHO) as low-grade (grade I), but 19.5% are considered high-grade (grade II or III) [1]. Up to 20% of meningiomas are clinically aggressive, regardless of histologic grading [2]. Complete surgical resection or radiotherapy are effective treatments for WHO grade I meningiomas, with an excellent prognosis with complete resection [2,3]. In many cases, however, there is disparity between histologic features and clinical behavior, creating difficulty in counseling patients on their prognosis and assessing the need for adjuvant treatment or more aggressive surveillance. Recently, molecular classifications have allowed us to better classify these tumors and better predict their behavior and the patient’s prognosis [4,5,6,7,8,9].

Several adjuvant therapeutic approaches have been used to treat tumors that can only be partially resected, aggressively recurrent tumors, or tumors in patients who are poor candidates for surgery or who do not want surgery. Stereotactic radiosurgery and fractionated radiotherapy are currently the adjuvant treatments of choice in the case of subtotal resection or in grade II/III meningiomas [3,10]. While improvement in progression-free survival following radiation has been reported for grade III meningiomas in some studies, the results are less clear for grade II lesions [11]. Furthermore, these therapies may raise the risk of secondary malignancies, radiation-induced brain necrosis, hypopituitarism, and cognitive disorders in up to 16.7% of cases [11,12].

Thus far, systemic therapy has been reserved for patients who cannot undergo surgery, patients with recurrent meningiomas, or those that are refractory to all surgical and radiotherapeutic treatments [12,13]. The safety and efficacy of systemic drugs, including chemotherapeutic, hormonal, and biologic agents, on refractory and high-grade meningiomas have been investigated for more than two decades [13,14]. In the present review, we discuss the available evidence regarding the safety and effectiveness of pharmacotherapeutic agents in treating meningiomas and review ongoing studies and future directions for medical therapy in the treatment of meningiomas.

## 2. Materials and Methods

### 2.1. Search Strategy

A search strategy was developed following the Preferred Reporting Items for Systematic Reviews and Meta-Analyses (PRISMA) guidelines [15]. This review is registered on the Open Science Framework website (https://osf.io/rvz8a (accessed on 10 December 2022)). The search term: (“Meningioma” [Mesh]) AND (“Antineoplastic Protocols” [Mesh]) OR (“Antineoplastic Agents” [Mesh]) OR (“Chemoradiotherapy” [Mesh]) was used to find the relevant articles on MEDLINE (www.pubmed.gov (accessed on 8 September 2022)) from the date of inception to September 2022. The resulting articles were reviewed for appropriateness for inclusion. The references of included articles were also hand-searched to avoid missing any relevant research endeavor. Next, the ClinicalTrials.gov (https://www.clinicaltrials.gov (accessed on 8 September 2022)) database of the National Library of Medicine was searched and screened in September 2022, using the search term “meningioma”. The “not yet recruiting”, “recruiting”, “active, but not recruiting”, and “enrolling by invitation” trials were selected by the website’s filters. The studies found through this search were screened for chemotherapy, hormone therapy, and biologic agent trials.

### 2.2. Selection Criteria

Studies found on PubMed were included if they: (1) were written in the English language; (2) were original prospective studies; and (3) evaluated the safety or efficacy of one or more pharmaceutical agent(s) for the treatment of meningioma. Studies were excluded if they: (1) conducted retrospective sub-group analyses and retrospective case series and case reports to mitigate the indirectness, selection bias, and reporting bias of the present study’s findings; (2) were congress abstracts and news articles; or (3) were reviews, meta-analyses, editorials, letters, or books.

### 2.3. Data Extraction

Extracted variables for included studies were: (1) investigator’s name; (2) journal and date of publication; (3) country of study; (4) eligibility and exclusion criteria; (5) participants’ characteristics (age, sex, performance status, previously received treatments); (6) method of diagnosis and grading of the meningioma; (7) follow-up examinations (e.g., routine clinic visits, laboratory evaluations, imaging); (8) the primary intervention; (9) outcomes of the study (e.g., radiographic response, survival, adverse events); (10) funding source; and (11) conflict of interest. The main outcomes of interest were radiographic response, overall survival, progression-free survival, six-month progression-free survival, and adverse events. Frequencies and percentages were utilized to present categorical variables. Means and ranges were used to present the continuous variables. The median of the continuous variable was reported in cases where the mean was unavailable. Meta-analysis of outcomes was not considered because of the limited number of included studies and outcome measure variability.

Extracted variables for ongoing studies chronicled in ClinicalTrials.gov were: (1) investigator’s name; (2) NCT number; (3) country of study; (4) funding source (public, private, or both); (5) primary intervention;(6) trial sites; (7) study phase; (8) enrollment target; (9) whether the study included only patients with meningioma; (10) study length; (11) recruitment status of the study; and (12) primary outcome of interest.

## 3. Results

The initial PubMed search resulted in 320 studies for initial review (Figure 1). Of these, 272 studies were excluded during title and abstract screening because of ineligible study type (n = 180), population (n = 35), outcomes (n = 27), and intervention (n = 7), or being written in a non-English language (n = 23). The full text of 48 papers was assessed for eligibility. Twenty-five papers were excluded due to retrospective design (n = 17), being non-clinical (n = 5), and having ineligible intervention (n = 1), population (n = 1), or outcomes (n = 1). One clinical trial was excluded since only one of the 21 patients was diagnosed with meningioma [16]. Overall, 23 studies were identified via MEDLINE search and 11 were identified through hand-searching of the references (Table 1 and Table 2). Initial search of the ClinicalTrials.gov portal yielded 78 trials for initial review. Of these, 51 studies were excluded because they did not involve therapeutic interventions (n = 25), were not drug-based interventions (n = 22), or did not include target patients with meningioma (n = 4). Ultimately, 27 ongoing clinical trials were identified as appropriate for inclusion in this study (Table 3).

### 3.1. Completed Clinical Trials

#### 3.1.1. Study Characteristics

Most of these clinical trials were performed in the USA (n = 22) over a 36-year span. The enrolled patients generally had meningiomas that were either unresectable, recurrent, or progressive despite all surgical and radiotherapeutic treatments based on clinical and neuroimaging evaluations. The patients were adults (>18) in all studies. The number of patients enrolled across all trials ranged from 4 to 164, with the median number of patients being 16. Most of the 740 trial participants were female (n = 474, 64.05%). Tumors were intracranial in all but two studies [24,30]. All but three were single-arm [42,43,46], and only one had a comparator group of standard care [42]. Only four studies focused on a particular meningioma grade [28,34,35,50], and all others included more than one grade. Hydroxyurea was the most frequently investigated chemotherapeutic agent, used in 12 studies (Table 4), followed by octreotide (n = 4), and mifepristone (n = 3).

#### 3.1.2. Outcomes

Overall, eight studies reported partial radiographic responses to the treatment [17,20,21,22,29,33,42,43]. No study reported a complete response. Administered regimens in these trials were: (1) oral tamoxifen, 30 mg divided by three doses a day [17]; (2) oral mifepristone, 200 mg in one daily dose [20,33,42]; (3) oral tamoxifen, 40 mg for four days and 10 mg thereafter divided by two doses per day [21]; (4) three or six one-month cycles of intravenous cyclophosphamide (500 mg/m^2^/day) and doxorubicin (15 mg/m^2^/day) on days 1–3 in addition to one dose of vincristine (mg/m^2^/day) within days 10–14 [22]; (5) oral hydroxyurea, 20 mg/kg/day [29]; and (6) 42-day cycles of oral sunitinib malate (50mg/day) on days 1–28 [43]. Notably, only one of these eight trials was a randomized clinical trial and this did not find the difference in partial radiographic response between the intervention and placebo groups to be significant [42]. All other trials were single-arm. Of the trials that reported partial radiological responses, two trials defined a partial radiographic response as any decrease in the largest diameter of the tumor [17,20] while the other six used MacDonald criteria (Table 5) [51] for radiographic response [21,22,29,33,42,43]. Time to tumor progression ranged from three to more than 957 weeks. Overall survival ranged from 22 days to more than nine years. When measured, six-month progression survival was 6–100%. Hematological disorders including leukopenia, anemia, and thrombocytopenia were the most common adverse events with more than 65 cases. Of the 18 pharmacotherapeutic regimens, only five had evidence of partial radiographic response (Table 4).

### 3.2. Ongoing Clinical Trials

#### 3.2.1. Status and Coordination

All ongoing clinical trials started between 2008 and 2022. Expected completion dates range from 2021 to 2029. Of the 27 included studies, 21 are occurring in the USA. The number of trial sites per study range from 1 to 705, with a median of one. Most trials are privately funded (16 of 27 or 59.2%). Four are funded privately and publicly through the National Institute of Health (NIH) or a foreign equivalent. Seven trials are completely publicly funded. Nineteen studies are currently recruiting patients, five are active but no longer recruiting, and three are not yet recruiting.

#### 3.2.2. Trial Design

Eight studies are phase I clinical trials, fourteen are phase II trials, two are phase I and II trials, and one is a phase II and III trial. No phase IV trials were found in the database. Two studies did not designate a trial phase. The target number of enrollment across all ongoing clinical trials ranges from 9 to 180, with a median number of patients being 34. Seventeen trials focus only on meningioma, with ten focusing on the drug of interest’s impact on other tumors as well. The most common primary outcome of interest is progression-free survival in eighteen trials, followed by toxicity in ten trials. Other primary outcomes of interest include drug pharmacokinetics, gene expression following drug administration, immunogenicity, changes in tumor size, and radiological response rates.

#### 3.2.3. Pharmacotherapy Targets

The conventional chemotherapy agents being investigated include gemcitabine, etoposide, ifosfamide, and carboplatin. A wide variety of targeted agents are also investigated, including mitogen-activated protein kinase (MEK)/mitogen-activated protein kinase (MAPK) pathway inhibitors (trametinib, selumetinib), phosphoinositide 3-kinase (PI3K)/protein kinase B (AKT)/the mammalian target of rapamycin (mTOR) pathway inhibitors (alpelisib, vistusertib), epidermal growth factor receptor (EGFR) inhibitors (brigatinib, afatinib), vascular endothelial growth factor receptor (VEGFR) inhibitors (cabozantinib, apatinib), c-MET and AXL inhibitor (cabozantinib), smoothened (SMO) inhibitors (sonidegib and vismodegib), focal adhesion kinase (FAK) inhibitor (GSK2256098), AKT inhibitor (capivasertib), cyclin-dependent kinase (CDK) inhibitors (ribociclib, abemaciclib), histone deacetylase inhibitor (OSU-HDAC42), glycogen synthase kinase 3 beta (GSK-3β) inhibitor (9-ING-41), and dopamine receptor D2 (DRD2) inhibitor (ONC206) (Figure 2). Biologic therapies being investigated in current clinical trials consist of PD-1 inhibitors (nivolumab, pembrolizumab, and sintilimab), PD-L1 inhibitors (avelumab), VEGF inhibitor (bevacizumab), and perillyl alcohol (NEO100) (Figure 3). Both hormone-based pharmacotherapies that are currently being investigated in clinical trials are targeted radionuclide therapies with radiolabeled somatostatin analogue (177Lu-DOTATE and 177LU-DOTA-JR11) (Figure 4).

## 4. Discussion

Classically, most meningiomas are WHO grade I lesions that can be cured surgically. This is one reason that advancements in medical treatments for aggressive recurrent or unresectable meningiomas have lagged behind treatments for other neoplasms. Physicians and scientists have been working to develop non-surgical therapeutic approaches for meningiomas for more than 35 years, with no effective pharmacotherapeutic agents currently recognized as standard of care. Moreover, thus far, all drug-based trials targeting meningioma have shown negative results. We previously performed multi-platform profiling of meningiomas and proposed three molecular groups based on transcriptional profiles that allowed us to better classify and prognosticate these tumors [7]. We demonstrated that these three groups could be identified using cytogenetics, DNA methylation profiling or transcriptional profiling [6]. Recently, several other groups have started to converge on the same biological groups [8,9,52]. As the field moves towards crystallizing a molecular classification scheme, looking through the lens of molecular groups may allow for new insights when both evaluating new pharmacotherapies and re-evaluating previously studied pharmacotherapies.

### 4.1. Completed Clinical Trials

While no clinical trial of biological agents has yet yielded a radiographic response, there were several chemotherapy and hormonal therapy regimens that were associated with a radiographic response. Overall, five of eighteen pharmacotherapeutic regimens demonstrated evidence of radiographic response, which were: (1) hydroxyurea; (2) cyclophosphamide, doxorubicin, and vincristine; (3) sunitinib; (4) mifepristone; and (5) tamoxifen.

Hydroxyurea is a ribonucleotide inhibitor that induces apoptosis by stopping the cell cycle in the S phase [23]. While one clinical trial showed partial radiographic response following treatment with this drug, other trials did not find any radiographic response following the therapy. Thus, it is hard to justify this drug’s limited efficacy in light of its more established hematological and dermatological side effects [23,24,26,27,29,30,31,34,47]. Hydroxyurea was also used as an adjunct to treatment with imatinib in two trials. However, no significant radiographic response to treatment in the two aforementioned studies was observed [39,46]. The same concerns apply to the combinatorial cytotoxic chemotherapy of cyclophosphamide/doxorubicin/vincristine, whose modest evidence of partial radiographic response in 21% of the patients was outweighed by much more pronounced evidence of dermatological and hematological adverse effects, which occurred in 100% of the patients [22,43].

The hormonal agents mifepristone and tamoxifen have both been studied in meningiomas. Mifepristone is an anti-progesterone drug that was found to result in partial radiographic in two earlier single-arm clinical trials [20,33]. However, its efficacy was similar to placebo with regard to radiographic response, time to tumor progression, overall survival and six-month progression-free survival in a phase III placebo-controlled clinical trial [42]. Tamoxifen is an anti-estrogen agent that was also associated with a radiographic response [17,21]. However, both clinical trials looking at tamoxifen were single-arm, and its effectiveness was not compared to a proper comparator group, limiting the impact of these findings.

There are several limitations to the methodology of the eight studies reporting the five regimens that demonstrate a radiographic response. First, two studies defined any reduction in tumor size as a response to treatment [17,20], which may lead to an overestimated response. Next, while the other six studies [21,22,29,33,42,43] used the established MacDonald radiographic response criteria [51], their results should be interpreted cautiously. Many patients in these clinical trials had undergone prior surgical resection or radiotherapy, which may significantly confound the noted response. Moreover, prior studies have shown meningioma aggressiveness to be significantly correlated with irregular shape, which may make measuring an adequate response difficult in patients with recurrent disease [16,53]. Furthermore, most of these studies had small sample sizes with limited statistical power and were single-arm, which inevitably introduces selection bias. The clinical validity of the results of these trails is further questioned by the fact that the only randomized placebo-controlled clinical trial of the eight studies did not demonstrate any significant difference in response to treatment between the systemic therapy and placebo [42].

### 4.2. On-Going Clinical Trials

Over the past three decades, several studies have chronicled the genomic makeup and common mutations found in meningioma [52,54,55,56,57,58,59,60]. Thus, there are several ongoing clinical trials of agents targeting molecular pathways known to contribute to the pathogenesis of meningiomas. For example, some meningiomas are found to have a highly expressed hedgehog signaling pathway, making somatic mutations of Smoothened (SMO) a potential target for chemotherapy [61,62,63]. Based on this, SMO inhibitors, sonidegib and vismodegib, are currently being investigated as a treatment for meningiomas (NCT03434262 and NCT02523014). Similarly, considerable activation of mutated AKTs in the PI3K/AKT/mTOR pathway is also reported in some meningiomas, and trials of inhibitors of these pathways (capivasertib, alpesilib, vistusertib) are also ongoing [61,62,63]. Another active signaling pathway in meningioma is the RAS/RAF/MEK/MAPK pathway, which transduces the VEGFR, EGFR and PDGFR’s pro-mitotic signals [61,64]. This signaling mechanism is currently the target of two distinct trials, using either trametinib or selumetinib (NCT03631953, NCT03095248). In addition to studying the efficacy of pharmacotherapies that target SMO (vismodegib), the PI3K/AKT/mTOR pathway (capivasertib), and CD4 and CD6 (abemacicilib), the ongoing multi-center Alliance trial (NCT02523014) is also studying the efficacy of the GSK2256098 FAK inhibitor. FAK is a cytoplasmic tyrosine kinase which integrates signals from integrins and growth factors to regulate cell proliferation, migration, and survival. Preclinical data have suggested increased vulnerability to FAK inhibitors in merlin-deficient tumor cells [65,66].

While some clinical trials have focused on molecular signaling pathways, others have utilized inhibition of certain growth factor receptors as a point of intervention. More than half of meningiomas exhibit overexpression of EGFR [63]. While a previous trial of EGFR inhibitors, gefitinib and erlotinib, did not result in any radiographic response to treatment [37], two ongoing trials of EGFR inhibitors, afatinib and brigatinib, are designed to investigate the safety and efficacy of these agents. As next-generation tyrosine kinase inhibitors, afatinib and brigatinib are effective against targets of gefitinib and erlotinib as well as less common EGFR mutations that might be resistant to gefitinib and erlotinib’s action (NCT02423525 and NCT04374305). Therefore, it is reasonable to evaluate whether these agents might be effective against meningiomas despite the failure of first-generation EGFR inhibitors. Another tyrosine kinase inhibitor of interest currently being investigated in two ongoing trials are VEGFR2 inhibitors, apatinib and carbozantinib (NCT04501705 and NCT05425004). Furthermore, a currently recruiting phase II trial (NCT02847559) and a recently completed phase II trial with pending results (NCT01125046) explored the impact of the VEGFR2 inhibitor bevacizumab in the treatment of patients with recurrent or progressive meningioma. While previous trials of VEGF receptor inhibitors sunitinib, vatalanib, and bevacizumab did not demonstrate any considerable radiographic response, they were efficacious in increasing six-month progression-free survival, showing that some benefit may be offered by antiangiogenic treatments in meningiomas [40,43,63,67,68]. Moreover, carbozantinib also inhibits receptor tyrosine kinases AXL and c-MET, which have both previously been shown to demonstrate elevated expression in recurrent meningiomas [69,70,71].

Targeting dysregulated cell growth is another strategy that is currently under investigation. It is known that the inactivation of cyclin-dependent kinase inhibitor 2A (CDKN2A) and CDKN2B genes leads to activation of CDK4 and CDK6, which may contribute to poorer outcomes and higher recurrence rates [63,72]. Two ongoing trials investigating ribociclib and one investigating abemaciclib as selective inhibitors of CDK4 and CDK6 are investigating the role of CDK inhibitors in treating meningiomas (NCT02933736, NCT03220646, NCT02523014). DNA hyper-acetylation by histone deacetylase inhibitor AR-42 has also shown promising anti-proliferative activity in preclinical meningioma models [4,73], which is now being used in the first clinical trial for the treatment of meningiomas (NCT05130866) [72]. The third agent in this category is 9-ING-41, an inhibitor of glycogen synthase kinase-3β (GSK-3β), which upregulates NF-κB’s transcriptional activity (NCT04239092) [63,74]. GSK-3β inhibition has shown promising results in the treatment of multiple malignancies, and is now being investigated for several refractory neoplasms, including meningiomas [75]. The last drug in this category is ONC206 (NCT04541082). This imipiridone small molecule increases the activity of TNF-related apoptosis-inducing ligand, which is a major contributor to the cytotoxicity of tumor cells [76]. Its safety and dose-escalation are currently being examined in a phase I trial on primary CNS neoplasms, including meningiomas (NCT04541082).

With regard to biologic agents, most ongoing clinical trials focus on immune checkpoint inhibition through PD-1 or PD-L1 inhibition. PD-L1-expressing tumor cells inhibit T-cell activation by binding to the PD-1 surface receptor on T- and B-cells [77] (Figure 3). In meningiomas, elevated expression of PD-L1 correlates with higher tumor grade and, subsequently, worse prognosis [78,79]. Moreover, tumors that have received prior radiation therapy also have significantly higher PD-L1 expression [78]. Given these findings, seven ongoing trials are exploring the effect of anti-PD1 or anti-PD-L1 therapy on patients with meningioma. A phase II trial is comparing the use of nivolumab PD-1 inhibitor alone to combination therapy with the anti-CTLA-4 antibody, ipilimumab (NCT02648997). A phase I trial is investigating the preoperative use of the PD-L1 inhibitor, avelumab, in combination with proton therapy for 3 months to evaluate its effect on the unresected tumor volume (NCT03267836). Other trials in this category target patients with high-grade meningioma who have failed surgical resection.

Hormone-based pharmacotherapy currently being explored in clinical trials centers around the somatostatin receptor-targeted radioactive drug, LUTATHERA (177 Lu-DOTATE). This drug binds to somatostatin receptors on tumor cells and delivers high doses of radiation. While a prior phase II clinical trial exploring the effects of this drug on patients with unresectable meningiomas did not demonstrate tumor regression with this treatment, it did find some efficacy when it came to slowing tumor progression [44]. A current phase I trial is investigating the safety and efficacy of a slightly modified drug, 177 Lu-DOTA-JR11, which has been shown to exert higher binding affinity for somatostatin receptors than 177 Lu-DOTATE and thus, postulated to have improved clinical efficacy and therapeutic index when treating advanced and recurrent meningiomas (NCT04997317) [80]. Two ongoing phase II trials are further elucidating the efficacy of LUTATHERA in treating high grade meningioma (NCT04082520 and NCT03971461). NCT04082520, in particular, is evaluating the efficacy of LUTATHERA in patients with progressive meningioma who have received external beam radiation therapy. One phase I and phase II trial is evaluating the safety and efficacy of LUTATHERA in pediatric (phase I) and young adult (phase 2) patients with progressive or recurrent high-grade CNS tumors including meningioma (NCT05278208).

### 4.3. Limitations

There are several limitations common to the studies presently available in the literature. All included studies had a small sample size, were underpowered, and had no “standard care” comparison group. When added with inadequate measures to mitigate the confounding factors, selection bias, information bias, or reporting bias, it is impossible to draw a robust conclusion from the current literature. Designing a medical therapy trial for meningiomas is a known challenge due to the scarcity of patients that do not respond to GTR and radiotherapy, the inevitably heterogeneous sample population with regard to the grade and natural history of the disease, difficulty in creating an acceptable set of outcomes in the short-term resulting in heterogeneity of outcome measures, and issues arising from long follow-up times because of the natural history of the disease [67,72]. Even when taking the results of these clinical trials into account, the evidence of effectiveness of these drugs on meningiomas is lacking. This is consistent with the findings of previous reviews on medical therapy and chemotherapy of meningiomas [23,67,68,72,81].

The present review is also limited by the following factors. Due to the heterogeneity of outcome measures and the scarcity of eligible trials, no meta-analysis could be conducted. Therefore, the risk of bias of these trials could not be quantitatively evaluated. Additionally, ClinicalTrials.gov, while a comprehensive database, is U.S.-based and thus, may not contain an exhaustive list of all clinical trials being conducted around the world. It is likely that some trials may not have been registered on this database.

## 5. Conclusions

While most meningiomas are relatively slow-growing and histologically benign, a subset of these tumors are aggressive and remain challenging to treat with the existing options of surgical resection and radiotherapy. No systemic therapeutics, thus far, have shown efficacy in the treatment of meningioma in the recurrent setting. Moreover, the heterogeneity of outcome measurements of existing clinical trials precludes a quantitative meta-analysis. Insights into the genomic and epigenomic make-up of meningiomas have provided new targets for potential systemic therapies. There are several ongoing clinical trials which act on molecular targets (SMO, AKT, FAK, etc.) previously studied in the preclinical setting. Furthermore, chemotherapies that dysregulate cell growth (CDK inhibitors and GSK-3β inhibitors) and induce apoptosis through caspase activity (DRD2 inhibtor) are also being studied in ongoing clinical trials. Finally, trials studying biologic therapies that prevent checkpoint inhibition (PD-1 inhibitors and PD-L1 inhibitors) and hormone therapy targeting somatostatin receptors with radioactive analogs also exhibit exciting potential as systemic pharmacotherapies for meningioma in the recurrent setting.

## Figures and Tables

**Figure 1 cancers-15-00483-f001:**
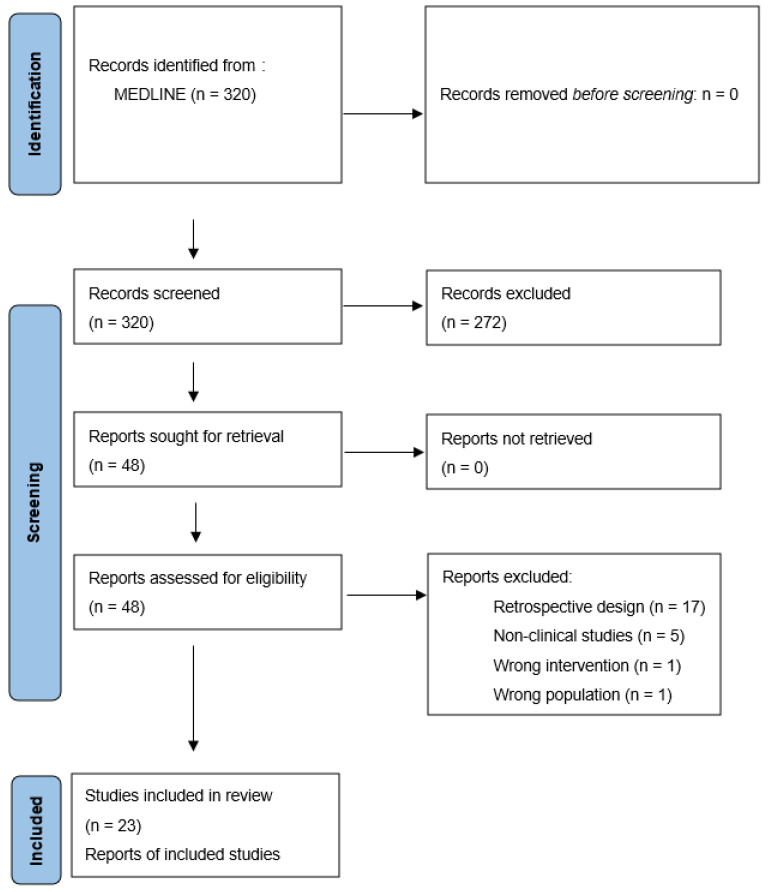
PRISMA flow diagram illustrating the study selection process on MEDLINE (www.pubmed.gov).

**Figure 2 cancers-15-00483-f002:**
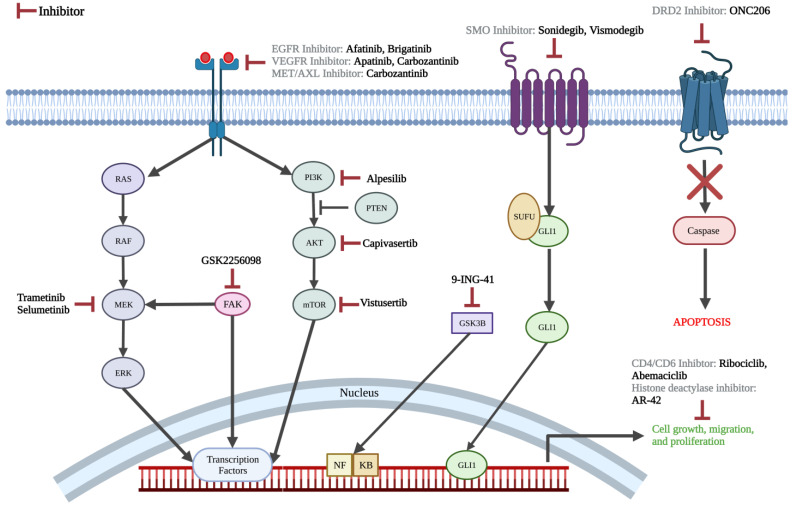
Molecular pathway highlighting the action of chemotherapy agents being studied in ongoing clinical trials. Made on biorender.com.

**Figure 3 cancers-15-00483-f003:**
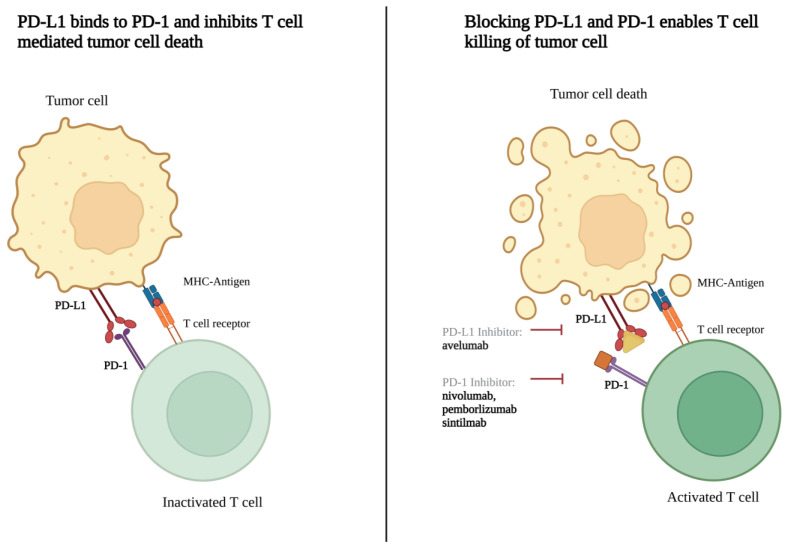
Mechanism of PD-1 to PD-L1 interaction inhibition. Made on biorender.com.

**Figure 4 cancers-15-00483-f004:**
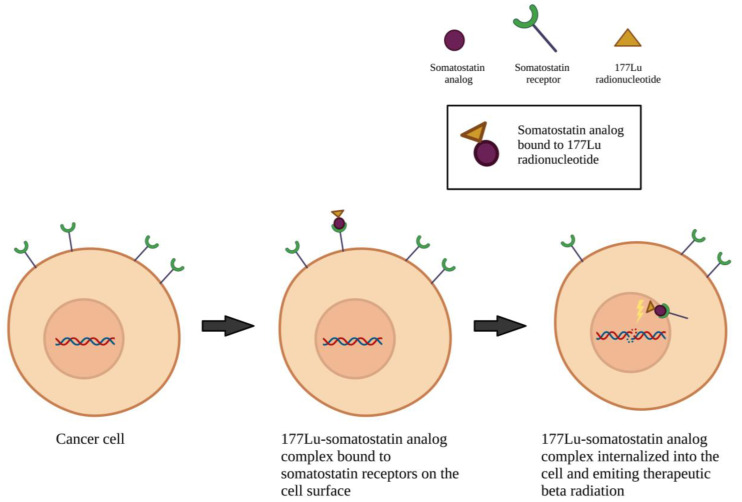
Mechanism of action of targeted radiolabeled somatostatin analogs. Made on biorender.com.

**Table 1 cancers-15-00483-t001:** Characteristics of the included studies.

Author, Year	Region	Participants	Meningioma Grade *	Age (mean)	KPS (Mean, Range)	Intervention
Markwalder et al., 1985 [17]	Switzerland	6 (5F/1M)	N/A	68.5y	N/A	Tamoxifen, 30 mg T.D.S.
Jääskeläinen et al., 1986 [18]	Finland	5 (5F)	I (n = 4) III (n = 1)	N/A	N/A	Medroxyprogestrone acetate, 1000 mg I.M. daily for 5 d and 1000 mg I.M. weekly thereafter
Grunberg et al., 1990 [19]	USA	9 (5F/4M)	N/A	41.55 y	Median: 70% (90–100)	Megestrol acetate, 160–320 mg Q.I.D.
Grunberg et al., 1991 [20]	USA	14 (8F/6M)	N/A	54.07 y	Median: 90% (60–100)	Mifepristone, 200 mg P.O. daily for min of 1 y
Goodwin et al., 1993 [21]	USA	19 (13F/6M)	N/A	Median: 58 y	N/A	Tamoxifen, 40 mg B.I.D. for 4 d and 10 mg B.I.D. thereafter
Chamberlain et al., 1996 [22]	USA	14 (6F/8M)	N/A	Median: 51 y	Median: 90% (70–100)	3 or 6 1-month cycles of cyclophosphamide (500 mg/m^2^/d, I.V. days 1–3), doxorubicin (15 mg/m^2^/d, days 1–3), and vincristine (1.4 mg/m^2^/d any day between days 10–14)
Schrell et al., 1997 [23]	Germany	4 (2F/2M)	N/A	48.25 y	All >70%	Hydroxyurea, 1000–1500 mg/day (approx. 20 mg/kg/d), daily P.O., for min of 2 y
Newton et al., 2000 [24]	USA	17 (13F/4M)	N/A	57.2 y	N/A	Hydroxyurea, 20 mg/kg/d (1250–1500 mg) P.O.
Muhr et al., 2001 [25]	Sweden	12 (8F/4M)	I (n = 6) II (n = 1) III (n = 3)	56 y	N/A	Interferon a, 1,500,000–5,000,000 IU S.C. daily
Mason et al., 2002 [26]	USA/Canada	20 (11F/9M)	I (n = 16) II (n = 3) III (n = 1)	Median: 59 y	80% (50–100)	Hydroxyurea, 1000–1500 mg/d (20–30 mg/kg/day) until clinical or imaging evidence of progression or 2 y
Rosenthal et al., 2002 [27]	Australia	15 (13F/2M)	I (n = 10) II (n = 5)	Median: 39 y	N/A	Hydroxyurea, 20 mg/kg/d daily P.O.
Chamberlain et al., 2004 [28]	USA	16 (11F/5M)	I (n = 16)	Median: 62.5 y	Median: 80% (60–100)	Temozolomide, 50–75 mg/m^2^/d P.O. daily for 42 days, 28 d drug holiday thereafter
Loven et al., 2004 [29]	Israel	12 (7F/5M)	I (n = 8) II (n = 4)	N/A	ECOG: grade I (n = 6), grade II (n = 4), grade III (n = 2)	Hydroxyurea, 20 mg/kg/d daily P.O. for 24 mo.
Newton et al., 2004 [30]	USA	21 (17F/4M)	I (n = 16) II (n = 1)	Median: 59 y	>60%	Hydroxyurea, 20 mg/kg/d P.O.
Hahn et al., 2005 [31]	Germany	21 (14F/7M)	I (n = 13) II (n = 2) III (n = 2)	Median: 60 y	N/A	Hydroxyurea, 1000–1500 mg (20 mg/kg/d) P.O.
Chamberlain et al., 2006 [32]	USA	16 (11F/5M)	N/A	Median: 60.5 y	Median: 80% (60–100)	Irinotecan, 350–600 mg/m^2^/d I.V. every 3 w for 9 w
Grunberg et al., 2006 [33]	USA	28 (19F/9M)	N/A	Median: 56 y	N/A	Mifepristone, 200 mg P.O. daily
Weston et al., 2006 [34]	UK	6 (F)	I (n = 5)	46 y	N/A	Hydroxyurea: starting at 15 mg/kg/d P.O. for 1 y
Chamberlain et al., 2008 [35]	USA	35 (29F/6M)	I (n = 35)	median: 61 y	Median: 80% (60–100)	Interferon-a, 10,000,000 U/m2 S.C. every other day
Wen et al., 2009 [36]	USA	23 (13F/10M)	I (n = 13) II (n = 5) III (n = 5)	Median: 58 y	Median: 80% (60–100)	Imatinib mesylate, 600–800 mg/d daily P.O. for 4 w cycles
Norden et al., 2009 [37]	USA	25 (13F/12M)	I (n = 8) II (n = 9) III (n = 8)	Median: 57 y	Median: 90% (60–100)	Gefitinib 500–1000 mg/d daily P.O. OR erlotinib 150 mg/d, daily P.O. in 4 w cycles
Johnson et al., 2011 [38]	USA	12 (3F/9M)	I (n = 3) II (n = 3) III (n = 6)	48.91y	All ECOG<3	Octreotide, 500 mcg S.C. T.D.S
Reardon et al., 2012 [39]	USA	21 (12F/9M)	I (n = 8) II (n = 9) III (n = 4)	Median: 51 y	Median: 80% (70–100)	Imatinib, 400–500 mg/d daily, hydroxyurea 1000 mg/day B. I. D.
Raizer et al., 2014 [40]	USA	25 (10F/15M)	I (n = 2) II (n = 14) III (n = 8)	Median: 59 y	Median: 80% (60–100)	Vatalanib, 500–1000 mg/d B. I. D. P.O. in 4 w cycles
Simo et al., 2014 [41]	Spain	9 (1F/8M)	II (n = 5) III (n = 4)	Median: 65 y	Median: 80% (60–100)	Octreotide LAR: 30–40 mg I.M. every 28 d
Ji et al., 2015 [42]	USA	164 (116F/48M) Intervention: 80 (57F/23M), comparator: 84 (59F/25M)	N/A	Median (intervention): 60.6 y Median (comparator): 53.2 y	All ECOG<3	Intervention: mifepristone, 200 mg P.O. daily for 2 y or disease progression Comparator: placebo
Kaley et al., 2015 [43]	USA	36 (22F/14M) EC: 13 (8F/5M)	I (n = 4) II (n = 30) III (n = 6)	Median: 61 y Median (EC): 48 y	Median: 80% (60–100) Median (EC): 90% (60–100)	Sunitinib malate, 50 mg/d, for days 1–28 of 42 d cycles
Marincek et al., 2015 [44]	Switzerland	34 (25F/9M)	N/A	Median: 61.3 y	N/A	90Y-DOTATOC and 177Lu-DOTATOC for 3 d every 6 (or more) w
Norden et al., 2015 [45]	USA	34 (17F/17M)	I (n = 16) II (n = 12) III (n = 6)	Median: 54 y	Median: 85% (60–100)	Octreotide LAR, 60 mg I.M. every 4 w
Mazza et al., 2016 [46]	Italy	15 (8F/7M) arm A, combinatorial intervention (n = 7) arm B, hydroxyurea alone (n = 8)	Arm A: I (n = 1) II (n = 4) III (n = 1) Arm B: I (n = 1) II (n = 5)	Median: Arm A = 68 y Arm B = 68.5y	Median ECOG: Arm A = 1 (0–2) Arm B = 1 (0–2)	Arm A: hydroxyurea, 1000 mg/d B. I. D. and imatinib, 400–600 mg/d daily Arm B: hydroxyurea, 1000 mg/d B. I. D.
Karsy et al., 2016 [47]	USA	7 (6F/1M)	I (n = 2) II (n = 5)	Median: 56 y	90–100%	Hydroxyurea, 20 mg/kg/d (1000–1500 mg/d) B. I. D. P.O. and verapamil, 120–480 mg/d B. I. D. P.O.
Shih et al., 2016 [48]	USA	17 (9F/8M)	I (n = 4) II (n = 7) III (n = 5)	Median: 59 y	Median ECOG: 1 (0–3)	Bevacizumab, 10 mg/kg I.V. on days 1/15 of 28 d cycles Everolumus, 10 mg P.O. on days 1/15 of 28 d cycles
Graillon et al., 2020 [49]	France	20 (11F/9M)	I (n = 2) II (n = 10) III (n = 8)	Median: 55 y	all 50% and higher	Octreotide LAR, 30 mg I.M. monthly for 1–3 y Everolimus, 10 mg P.O. daily for 1–3 y
Karajannis et al., 2021 [50]	USA	8 (5F/3M)	I (n = 8)	43.12 y	all 60% and higher	Everolimus, 10 mg P.O. daily for 10 d preoperatively

* Based on WHO grading of meningiomas. KPS, Karnofsky Performance Scale; y, year(s); F, female; M, male; T.D.S., three times a day; I.M., intramuscular; Q.I.D., four times a day; d, day(s); I.V., intravenous; mo., month(s); min, minimum; mcg, micrograms; approx., approximately; P.O., per os; ECOG, Eastern Cooperative Oncology Group; E.C., exploratory cohort; LAR, long-acting releasing.

**Table 2 cancers-15-00483-t002:** Outcomes of the included studies.

Author	Partial/Complete Radiographic Response	Stable Radiographic Response	Time to Tumor Progression (Median, Range)	Overall Survival (Median, Range)	6-Month Progression-Free Survival	Grade III/IV/V Toxicities *
Markwalder et al., 1985 [17]	16.66%	50%	16 mo. (8–24)	24 mo.	100%	N/A
Jääskeläinen et al., 1986 [18]	0	N/A	Grade I: 21–45 mo. Grade III: 8 w	N/A	80%	N/A
Grunberg et al., 1990 [19]	0	N/A	N/A	N/A	N/A	N/A
Grunberg et al., 1991 [20]	30.76%	38.46%	5.33 mo. (2–8)	N/A	N/A	N/A
Goodwin et al., 1993 [21]	5%	32%	15.1 mo.	N/A	N/A	N/A
Chamberlain et al., 1996 [22]	21%	79%	4.6 y (2.2–7.1)	5.3 y (2.6–7.6)	N/A	N/A
Schrell et al., 1997 [23]	N/A	N/A	N/A	N/A	N/A	N/A
Newton et al., 2000 [24]	0	88%	80 w (20 -> 144)	N/A	N/A	Hematological (n = 5)
Muhr et al., 2001 [25]	N/A	N/A	N/A	N/A	N/A	N/A
Mason et al., 2002 [26]	N/A	N/A	Grade I: 54 w (41–66) Grade II: 25.33 w (12–45) Grade III: 24 w	N/A	N/A	Hematological (n = 3)
Rosenthal et al., 2002 [27]	0	85%	N/A	N/A	N/A	Hematological (n = 1), dermatological (n = 1)
Chamberlain et al., 2004 [28]	0	81.25%	5 mo. (2.5–5)	7.5 mo. (95%CI 7–8)	N/A	Hematological (n = 22), constitutional (n = 3), neurological (n = 1)
Loven et al., 2004 [29]	10%	0	13 mo. (4–24)	N/A	N/A	Hematological (n = 4), dermatological (n = 1)
Newton et al., 2004 [30]	0	90%	Mean: 176 w (20 -> 328)	N/A	N/A	Hematological (n = 6)
Hahn et al., 2005 [31]	0	71.5%	59 w (10–175)	N/A	N/A	N/A
Chamberlain et al., 2006 [32]	0	81%	4.5 mo. (2.25–10.5)	7 mo. (95% CI: 7–8)	6%	GI and hematological (n = 12), neutropenic fever (n = 1)
Grunberg et al., 2006 [33]	17.85%	N/A	N/A	N/A	N/A	N/A
Weston et al., 2006 [34]	0	50%	N/A	N/A	N/A	N/A
Chamberlain et al., 2008 [35]	0	74.28%	7 mo. (2–24)	8 mo. (3–28)	54%	fatigue (n = 6), hematological (n = 7), GI (n = 1)
Wen et al., 2009 [36]	0	47.36%	2 mo. (0.7–34)	N/A	29.4%	Hematological (n = 4), fluid and electrolyte (n = 3), other (n = 3)
Norden et al., 2009 [37]	0	32%	10 w (95% CI 8–20)	23 mo.	28%	22
Johnson et al., 2011 [38]	0	75%	17 w (3 -> 957)	2.7 y (22 d–9.7 y)	33.33%	N/A
Reardon et al., 2012 [39]	0	66.66%	7.0 mo. (95% CI 3.8–9.2)	66.0 mo. (95% CI 20.7, 66.0)	61.9% (95% CI 38.1–78.8)	Hematological (n = 3), other (n = 3)
Raizer et al., 2014 [40]	N/A	N/A	grade II: 6.5 mo. grade 3: 3.6 mo.	Grade II: 26.0 mo. Grade 3: 23 mo.	Grade II: 64.3% Grade III: 37.5%	Hepatic (n = 5), constitutional (n = 4)
Simo et al., 2014 [41]	0	33.33%	4.23 mo. (1–9.4)	18.7 mo. (2.7–39.9)	44.4%	0
Ji et al., 2015 [42]	Intervention: 1.4% Comparator: 1%	Intervention: 55% Comparator: 52%	Intervention: 10 mo. (95% CI 7–13) Comparator: 11 mo. (95% CI 6–18)	Intervention: 8 y Comparator: 12 y	N/A	Intervention: n = 37 Comparator: n = 25
Kaley et al., 2015 [43]	5.55%	69.44%	5.2 mo. (95% CI: 2.8–8.3)	24.6 mo. (95% CI: 16.5–38.4)	42%	CNS hemorrhage (n = 3), thrombotic microangiopathy (n = 2), GI (n = 1)
Marincek et al., 2015 [44]	0	65.6%	N/A	Mean: 8.6 y	N/A	Hematological (n = 3), renal (n = 1)
Norden et al., 2015 [45]	0	75%	Grade I: 26 w (12–43) Grade II/III: 15 w (8–20)	Grade I: N/A Grade II/III: 104 w (77–158)	32%	Metabolic (n = 10), fatigue (n = 2), other (n = 4)
Mazza et al., 2016 [46]	0	Arm A: 57.14% Arm B: 100%	Arm A: 4 mo. Arm B: 19 mo.	Arm A: 6 mo. Arm B: 27.5 mo.	N/A	Neurological (n = 2), GI (n = 1), hematological (n = 1)
Karsy et al., 2016 [47]	0	N/A	8.0 mo. (95% CI 6.1–9.9)	30.0 mo. (95% CI 22.8–37.2)	85% (95% CI 5.5–97.0%)	Hematological (n = 5), other (n = 1)
Shih et al., 2016 [48]	0	88%	22 mo. (95% CI 4.5–26.8)	23.85 mo. (95% CI 9–33.1)	69%	Hematological (n = 1), metabolic (n = 2), renal (n = 2), GI (n = 2), other (n = 3)
Graillon et al., 2020 [49]	0	N/A	6.6 mo. (95% CI 2.7–15)	N/A	55% (95% CI 31.3–73.5)	Stomatitis (n = 3), other (n = 5)
Karajannis et al., 2021 [50]	N/A	N/A	N/A	N/A	N/A	0

* Based on National Cancer Institute Common Toxicity Criteria and Common Terminology Criteria for Adverse events. Abbreviations: y, year(s); mo., month(s); w, week(s); CI, confidence intervals; GI, gastrointestinal.

**Table 3 cancers-15-00483-t003:** Characteristics of ongoing pharmacotherapy clinical trials.

Investigator	NCT Number	Region	Funding	Intervention	Trial Sites	Phase	Target Enrollment	Meningioma Only	Study Length	Recruitment status	Primary Outcome of Interest
** *Chemotherapy* **											
Scott R. Plotkin	03071874	USA	Both	AZD2014 (vistusertib), PO BID on days 2 and 7; 28-day cycles	3	2	25	Yes	2017–2024	Active, not recruiting	PFS
Thomas Graillon	03631953	France	Public	Trametinib (1.5 mg/d daily), Alpelisib (120–200 mg/d daily)	1	1	25	Yes	2019–2022	Recruiting	Toxicity
Jun-ping Zhang	04501705	China	Private	Apatinib Mesylate, 500 mg PO daily, until progressive disease; 28-day cycles	1	N/A	29	Yes	2020–2025	Recruiting	PFS
Nader Sanai	02933736	USA	Private	Ribociclib (LEE011), 900 mg PO QD; total of 5 doses before surgery	1	1	48	No	2016–2022	Recruiting	Pharmacokinetics, Toxicity
Rupesh Kotecha	05425004	USA	Private	Cabozantinib 60 mg QD for 28 days	1	2	24	Yes	2022–2024	Recruiting	PFS
NA	05130866	NA	Private	AR-42 (OSU-HDAC42) 30–60 mg 3 times a wk followed by 1 wk off	NA	2/3	89	Yes	2021–2027	Not yet recruiting	PFS
Ludimila Cavalcante	04239092	USA	Private	9-ING-41 (9.3 mg/kg IV twice per week), w. or w/o irinotecan (50 mg/m^2^/d IV days 1–5 of 21 d cycles)	8	1	48	No	2020–2023	Recruiting	Toxicity
Scott Plotkin	04374305	USA	Private	Brigatinib, PO daily	4	2	80	No	2020–2029	Recruiting	PFS
Trent Hummel	03095248	USA	Private	Selumetinib, 75 mg/d PO BID; 28 d cycles. up to 2 y	1	2	34	No	2017–2023	Recruiting	PFS
Mark Gilbert	04541082	USA	Both	ONC206 (imipridone class of anti-cancer small molecules)	1	1	102	No	2020–2024	Recruiting	Toxicity
Giles W. Robinson	03434262	USA	Private	Gemcitabine (IV), ribociclib (PO), sonidegib (PO), trametinib (PO), G-CSF (SC)	1	1	108	No	2018–2025	Recruiting	Pharmacokinetics, Toxicity
Priscilla Brastianos	02523014	USA	Both	A: Vismodegib PO Q.D.; 28 d cycles B: GSK2256098 PO QD; 28 d cycles C: Capivasertib PO BID; days 1–4 of 7 D: Abemaciclib PO BID; 28 d cycles	705	2	124	Yes	2015–2024	Recruiting	PFS
Thomas Kaley	03220646	USA	Private	Abemaciclib, 200 mg PO BID; 28 d cycles	8	2	78	No	2017–2023	Recruiting	PFS, RRR
Santosh Kesari	02423525	USA	Private	Afatinib, 80–280 mg, PO every 4 d or 7 d	1	1	24	No	2016–2021	Active, not recruiting	Toxicity
** *Hormone Therapy* **											
Dominik Cordier	04997317	Switzerland	Public	A: IV 4.5 GBq 177Lu-DOTA-JR11 (300–1300 ug) once; 2nd cycle of 200 ug B: IV 4.5 GBq 177Lu-DOTA-JR11 (300–1300 ug) once; 2nd cycle of 200 ug	1	1	18	Yes	2021–2025	Recruiting	Toxicity
Kenneth Merrell	04082520	USA	Public	IV Ga 68-DOTATE and then IV 177Lu-DOTA over 30–40 min. Cycles repeat every 8 wk for up to 6 mo.	1	2	41	Yes	2019–2024	Recruiting	PFS
Ralph Salloum	05278208	USA	Public	IV 177Lu-DOTATE (200 mCi) once every 8 wk for 8 mo.	9	1/2	65	No	2022–2026	Not yet recruiting	Toxicity PFS
Erik Sulman	03971461	USA	Private	IV 177Lu-DOTATE every 8 wk for 4 doses	2	2	32	Yes	2019–2023	Recruiting	PFS
** *Biological Therapy* **											
David Reardon	02648997	USA	Private	A. nivolumab, 240 mg q2w. B. EBRT followed by 4 cycles of nivolumab (2mg/kg q3w) + ipilimumab (1 mg/kg every 3 weeks) followed by nivolumab (480 mg q4w)	1	2	50	Yes	2016–2023	Recruiting	PFS
Priscilla Brastianos	03279692	USA	Private	IV pembrolizumab q3w	2	2	26	Yes	2017–2021	Active, not recruiting	PFS
Feng Chen	04728568	China	Public	IV sintilimab q3w	1	NA	15	Yes	2020–2023	Recruiting	PFS
Jiayi Huang	03267836	USA	Private	IV avelumab (10 mg/kg) and proton therapy (30 CGE) q2w for 3 mo. Surgical evaluation at 3 mo.	1	1	9	Yes	2017–2023	Active, not recruiting	Immunogenicity Tumor size
Jiayi Huang	03604978	USA	Public	A. IV nivolumab; 28 d cycles for 1 y. Multi-fraction stereotactic radiosurgery (days 1,3, and 5) B. IV nivolumab q2w for 6 mo. followed by q4w for 6 mo. IV ipilimumab q6w for 6 mo. multi-fraction stereotactic radiosurgery (days 1,3, and 5)	22	1/2	15	Yes	2018–2022	Active, not recruiting	Toxicity RRR
Nancy Bush	04659811	USA	Private	IV pembrolizumab 200 mg q3w with SRS	1	2	90	Yes	2020–2024	Recruiting	PFS
Marta Penas-Prado	03173950	USA	Public	IV nivolumab 240 mg q2w for 2 cycles and then 480 mg q4w for a total 14 doses	7	2	180	No	2017–2023	Recruiting	PFS
NA	05023018	NA	Private	NEO100 self-administered qid on 28 d cycle for up to 12 cycles	NA	2	30	Yes	2021–2026	Not yet recruiting	Toxicity PFS
Priya Kumthekar	02847559	USA	Both	IV bevacizumab q2w for 4 cycles and then q2w or q3w + daily electric field therapy with Optune device	9	2	27	Yes	2016–2024	Recruiting	PFS

PO., per os; d, day(s); wk, week(s); IV, intravenous; w., with; w/o, without; y, year(s); SC, subcutaneous; PFS, Progression Free Survival; RRR, Radiological Response Rate.

**Table 4 cancers-15-00483-t004:** Characteristics of studied pharmacotherapeutic regimens.

Agent	Dosage	Partial Radiographic Response (Range)	Stable Radiographic Response (Range)	6-Month Progression-Free Survival (Range)	Common Grade III/IV/V Toxicities
** *Chemotherapy* **					
Hydroxyurea	15–30 mg/kg/d 1000–1500 mg/d	0–10%	0–88%	85%	Hematological, dermatological
Temozolomide	50–75 mg/m^2^/d	0	81.25%	N/A	Hematological, constitutional, neurological
Irinotecan	350–600 mg/m^2^/d	0	81%	6%	GI/hematological
Cyclophosphamide Doxorubicin Vincristine	500 mg/m^2^/d 15 mg/m^2^/d 1.4 mg/m^2^/d	21%	79%	N/A	N/A
Imatinib	500–800 mg/d	0	47.3–100%	N/A	Hematological, fluid and electrolytes
Gefitinib Erlotinib	500–1000 mg/d 150 mg/d	0	32%	28%	N/A
Vatalanib	500–1000 mg/d	N/A	N/A	37.5–64.3%	Hepatic, constitutional
Sunitinib	50 mg/d	5.55%	69.4%	42%	Hemorrhagic/thrombotic events
Hydroxyurea Imatinib	1000 mg/d 400–600 mg/d	0	57.1–66.6%	61.9%	Hematological
Everolimus	10 mg/d for 10 d preoperatively	N/A	N/A	N/A	None
** *Hormonal therapy* **					
Octreotide	500 mg/d (regular) 30–60 mg monthly (LAR)	0	33.33–75%	32–44.4%	Hematological, metabolic, constitutional
90Y-DOTATOC 177Lu-DOTATOC	3 d every 6 or more wk	0	65.6%	N/A	Hematological, renal
Mifepristone	200 mg/d	1.4–30.76%	38.46–55%	N/A	N/A
Tamoxifen	10–30 mg/d	5–16.66%	32–50%	100%	N/A
Medroxyprogestrone acetate	1000 mg weekly	0	N/A	80%	N/A
Megestrol acetate	160–320 mg/d	0	N/A	N/A	N/A
** *Biologic agents* **					
Interferon a	1,500,000–5,000,000 IU/d 10,000,000 U/m^2^ every other day	0	74.28%	54%	N/A
** *Combined regimens* **					
Bevacizumab Everolumus, P.O. on days 1/15 of 28d cycles	10 mg/kg 10 mg/d	0	88%	69%	Hematological, metabolic, renal, GI
Octreotide LAR Everolimus	30 mg monthly 10 mg/d	0	N/A	55%	Stomatitis

Dd, day; wk, week(s); G.I., gastrointestinal; LAR, long-acting releasing.

**Table 5 cancers-15-00483-t005:** MacDonald criteria for assessment of brain tumor treatment response *.

Types of Response	Definition
Complete Response (CR)	Complete disappearance of all enhancing tumors on consecutive CT or MRI scans at least 1 month apart, off steroids, and neurologically stable or improved.
Partial Response (PR)	≥50% reduction in size of enhancing tumor on consecutive CT or MRI scans at least 1 month apart, steroids stable or reduced, and neurologically stable or improved
Progressive Disease (PD)	>25% increase in size of enhancing tumor or any new tumor on CT or MRI scans, or neurologically worse, and steroids stable or increased
Stable Disease (SD)	Imaging features do not qualify for CR, PR, or PD and neurologically stable

* Measurements are obtained at the tumor’s largest cross-sectional area.

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
