# Peer review of "The Role of Pharmacotherapy in Treatment of Meningioma: A Systematic Review"

_cancers, 2023, doi:10.3390/cancers15020483_

Round 1

Reviewer 1 Report (Previous Reviewer 1)

Similar comments than the previous version.

The authors performed a great work to present and synthetize these data. Text and tables are clear and well-presented.

Please, minor changes are required before acceptance:

-Jean-Olivier Arnaud is not the PI in table 3; Thomas Graillon is...

-Brastianos not Brastinos p15

Author Response

Similar comments than the previous version.

The authors performed a great work to present and synthetize these data. Text and tables are clear and well-presented.

Thank you for your time and for agreeing to review this manuscript. We are pleased to read your encouraging comments.

Please, minor changes are required before acceptance:

-Jean-Olivier Arnaud is not the PI in table 3; Thomas Graillon is...

Thank you for making this point, we have corrected Table 3.

-Brastianos not Brastinos p15

Thank you for pointing this out, we have corrected it.

Reviewer 2 Report (Previous Reviewer 2)

The authors have addressed the various comments.

In response to their queries:

a) the bev+TTF trial (NCT02847559) remains active and is undergoing continued accrual. 

b) The CDK4/6 inhibitor trial is undergoing development via NRG Oncology. In turn, it would be OK not to mention the specific trial in the manuscript.

Author Response

The authors have addressed the various comments.

Thank you for your time and for agreeing to review this manuscript.

In response to their queries:

  1. the bev+TTF trial (NCT02847559) remains active and is undergoing continued accrual. 

Thank you for this point, we have added it to the manuscript.

  1. b) The CDK4/6 inhibitor trial is undergoing development via NRG Oncology. In turn, it would be OK not to mention the specific trial in the manuscript.

Thank you for pointing this out. The aforementioned trial is not mentioned in the manuscript. On the other hand, CDK4/6 inhibitors are discussed in the Discussion section.

This manuscript is a resubmission of an earlier submission. The following is a list of the peer review reports and author responses from that submission.

Round 1

Reviewer 1 Report

The authors performed a descriptive review of previous prospective studies on systemic therapies for meningiomas and on ongoing clinical trials for meningiomas. The purpose of the review remains unclear. The review is complete but does not bring new data compared to previous articles on this topic. Some retrospective studies could also be of interest.Tables are clear and informative and the text is concise.

Reviewer 2 Report

The authors present a systematic review of systemic therapies for meningiomas.  The methodology for conducting the article search is fairly sound, although some well known studies (Alliance; NCT02523014) were omitted.  This raises  concern that more limited studies may have been missed as well.  My specific comments are detailed below.

1) In the Abstract, I would remove the comment that 8 studies demonstrated PR.  This is a bit misleading as the studies did not meet their primary efficacy endpoints.

2) Introduction, 1st paragraph, I would advocate adding the most recent CBTRUS epidemiologic data (Ostrum, et al. Neuro Oncol).

3) I would avoid using the terms "benign", "atypical", and "malignant" and would rely on the WHO grade instead.

4) Introduction, 1st paragraph, when discussing the molecular characterization, recent work on DNA methylation profiling (Choudhury A, et al. Nat Genet. 2022;54(5):649-659) should be included.

5) Table 3, the ongoing Alliance trial of targeted therapy and another trial of bevacizumab+TTF (NCT01125046)  are not listed.

6) Outcomes, Macdonald criteria should be explained, as they were initially intended for gliomas (although subsequently used for other tumors).

7) Pharmacotherapy targets, the Abs should all be lower case.

8) Discussion, 1st paragraph, I would argue that the results have not been conflicting.  The trials have all universally been negative.  There is consensus in the neuro-oncology community on this point.

9) Discussion, 1st paragraph, last sentence, while it will be important to consider previous trial results through the lens of more contemporary (yet to be validated) molecular subclassification, it will be unlikely that this will be more than a theoretical contemplation due to lack of adequate archival tissue for reanalysis of completed trials.

10) Completed Clinical Trials, it would be worthwhile to dig down into the methodology for defining response in all of the studies which yielded >0% response rate.  In routine clinical practice w/ these agents off-label, I suspect that most in the neuro-oncology community at best only observe stable disease.  It is interesting that agents with radiographic responses reported in almost 1/3 of patients (in a disease with dismally low response rates) were not able to meet their other study efficacy endpoints.

11) Ongoing clinical trials, 1st paragraph, the NCI cooperative group Alliance trial should be included in these discussions.  Ideally, the preclinical data (Brastianos, Nat Genet. 2013, and others) should be cited.

12) When discussing CDK4/6 the NRG Oncology cooperative group trial using RT+ribociclib should be mentioned.

13) Conclusions, this section has a boilerplate sentence and no actual content.  This should be added.